# Negative Correlation of Serum Adiponectin Level with Aortic Stiffness in Elderly Diabetic Persons

**DOI:** 10.3390/ijerph19053062

**Published:** 2022-03-05

**Authors:** Jui-Jung Hung, Du-An Wu, Ming-Chun Chen, Bang-Gee Hsu

**Affiliations:** 1School of Medicine, Tzu Chi University, Hualien 97004, Taiwan; 103311130@gms.tcu.edu.tw (J.-J.H.); despdu@yahoo.com.tw (D.-A.W.); 2Division of Metabolism and Endocrinology, Hualien Tzu Chi Hospital, Buddhist Tzu Chi Medical Foundation, Hualien 97004, Taiwan; 3Department of Pediatrics, Hualien Tzu Chi Hospital, Buddhist Tzu Chi Medical Foundation, Hualien 97004, Taiwan; 4Division of Nephrology, Hualien Tzu Chi Hospital, Buddhist Tzu Chi Medical Foundation, Hualien 97004, Taiwan

**Keywords:** adiponectin, aging, aortic stiffness, diabetes mellitus, carotid–femoral pulse wave velocity

## Abstract

Adiponectin has anti-inflammatory activity against atherosclerosis. Aortic stiffness is a common manifestation of atherosclerosis in diabetes mellitus and elderly persons. This study aimed to evaluate whether low serum adiponectin levels were associated with aortic stiffness in geriatric diabetic patients. Blood samples were obtained from 130 diabetic participants aged ≥ 65 years. We defined high aortic stiffness based on a carotid–femoral pulse wave velocity (cfPWV) of >10 m/s. Circulating adiponectin concentrations were examined using enzyme-linked immunosorbent assays. Sixty-six participants (50.8%) had aortic stiffness. Patients with aortic stiffness had lower serum adiponectin concentrations than those in the control group (*p* < 0.001). Multivariate logistic regression analysis showed that the adiponectin level (odds ratio: 0.939, 95% confidence interval: 0.898–0.981, *p* = 0.005) was an independent predictor of aortic stiffness in elderly diabetic persons. Multivariate forward stepwise linear regression analysis also demonstrated that the adiponectin level (β = −0.256, adjusted R^2^ change = 0.100, *p* = 0.003) was negatively associated with cfPWV values in older diabetic patients. In conclusion, serum adiponectin is negatively correlated with cfPWV and is an independent predictor of aortic stiffness in elderly diabetic persons.

## 1. Introduction

Diabetes mellitus (DM), a chronic metabolic disorder encompassing insulin resistance and hyperglycemia, remains a global healthcare burden [1]. There were 451 million adults diagnosed with DM in 2017, with an increasing prevalence of nearly 693 million by 2045 worldwide based on the prediction from the International Diabetes Federation [2]. According to the Framingham risk assessment study, DM increases the frequency of cardiovascular events. At a given level of baseline risk in patients without diabetes, the risk for cardiovascular events increases because of DM by approximately 10 years. Moreover, at any given level of other major risk factors, DM increases the risk by three to four times [3].

Arterial stiffness, a pathological condition with low-grade inflammation and abnormal regulation of collagen, elastin fibers, and cellular elements, represents subclinical organ damage and future cardiovascular disease (CVD) development [4,5]. Aortic stiffness, evaluated via the carotid–femoral pulse wave velocity (cfPWV), has been proven to be an essential predictor of future CVD or all-cause mortality, independent of classic cardiovascular risks [4,6]. Adiponectin, the most abundant peptide secreted by white adipocytes, protects against the development of obesity-related diseases [7]. Adiponectin, by preventing oxidative stress, insulin resistance, and inflammatory processes and alleviating the development of atherosclerosis via pleiotropic modulations of multiple vascular cell types, plays a central role in CVD protection [8,9].

Previous studies revealed that hypoadiponectinemia had negative effects on arterial stiffness in young, normotensive patients with type 1 or type 2 DM [10,11]. However, the correlation between aortic stiffness and circulating adiponectin concentrations in elderly patients with type 2 DM remains unknown. To address this topic, this study aimed to investigate clinical and biochemical variables and the relationship between serum adiponectin concentrations and aortic stiffness in geriatric patients with type 2 DM.

## 2. Materials and Methods

### 2.1. Participants

The Research Ethics Committee of Hualien Tzu Chi Hospital, Buddhist Tzu Chi Medical Foundation, approved this study (IRB103-136-B). A total of 130 elderly patients with type 2 DM (≥65 years of age) were enrolled in this cross-sectional study conducted from November 2014 to March 2015 in a medical center in Hualien, Taiwan. All patients included in this study provided informed consent prior to enrollment. DM was defined as a fasting plasma glucose level exceeding 126 mg/dL or a patient using oral hypoglycemic medications or insulin [12]. Each participant’s blood pressure (BP) was measured using standard mercury sphygmomanometers after sitting for at least 10 min in the morning. After measuring the systolic blood pressure (SBP) and diastolic blood pressure (DBP) three times at 5 min intervals, the average value was taken for analysis. We defined hypertension as SBP ≥ 140 mmHg and/or DBP ≥ 90 mmHg or receiving any anti-hypertensive medication in the past 2 weeks, according to 2018 European Society of Cardiology and the European Society of Hypertension guidelines for the management of arterial hypertension [13]. Participants who had type 1 DM, heart failure, malignancy, or acute infection at the time of blood sampling and those who refused to provide informed consent were excluded.

### 2.2. Anthropometric Analysis

While wearing light clothing without shoes, participants were weighed to the nearest 0.5 kg, and their height was measured to the nearest 0.5 cm. Body mass index (BMI) was calculated by dividing the weight (kg) by the height (m) squared. The waist circumference at the shortest point between the lower edge of the rib cage and the iliac crest was measured [14,15].

### 2.3. Biochemical Investigations

Overnight fasting blood samples containing approximately 5 mL were immediately centrifuged at 3000× *g* for 10 min. Serum levels of blood urea nitrogen (BUN), creatinine, fasting glucose, glycated hemoglobin (HbA1c), total cholesterol, triglycerides (TG), and low-density lipoprotein cholesterol (LDL-C), as well as urine protein-to-creatinine ratio (UPCR) using random spot urine testing, were measured using an autoanalyzer (Siemens Advia 1800, Siemens Healthcare GmbH, Erlangen, Germany) [14,15]. The CKD-EPI (Chronic Kidney Disease Epidemiology Collaboration) equation was used for calculating the estimated glomerular filtration rate (eGFR) in this study. A commercially available enzyme immunoassay (SPI-BIO, Montigny le Bretonneux, France) was used to quantify serum adiponectin [14,15]. The intra-assay and inter-assay coefficients of variation in the measurement for adiponectin were 6.4% and 7.3%, respectively.

### 2.4. Aortic Stiffness by cfPWV Measurements

Aortic stiffness was evaluated by measuring the cfPWV values using pressure applanation tonometry (SphygmoCor system, AtCor Medical, Sydney, Australia), as previously described [11,14,15,16,17]. After a minimum of a 10 min rest in the morning, measurements were performed for all participants while in the supine position in a temperature-controlled room. The recording was performed simultaneously with the electrocardiogram (ECG) signal, which provided the *R*-timing reference. We performed continuous pulse wave recording at two superficial artery sites (carotid-femoral segment). By subtracting the distance from the carotid measurement point to the sternal notch from the distance from the sternal notch to the femoral measurement site, the carotid–femoral distance was obtained. We use integration software to process each set of pulse wave and ECG data to calculate the mean time difference between the *R*-wave and pulse wave on a beat-to-beat basis, with an average of 10 consecutive cardiac cycles. The distance and mean time difference between the two recorded points were used to calculate the cfPWV. The quality indicators included in the software are designed to ensure data consistency. We defined aortic stiffness as the cfPWV values > 10 m/s, and those whose values were ≤10 m/s were included in the control group in our study, according to the guidelines published by the European Society of Cardiology and the European Society of Hypertension [13].

### 2.5. Statistical Analysis

Data were tested for normal distribution using the Kolmogorov–Smirnov test. Data distributed normally were compared using Student’s independent *t*-test (two-tailed) and are presented as the mean ± standard deviation. Non-normally distributed data are represented by the median and interquartile range, and comparisons between participants were examined via the Mann–Whitney U test (age, TG, fasting glucose, BUN, creatinine, and UPCR). The chi-square test was used to analyze data expressed as the number of patients. Multivariate logistic regression analysis was performed to test the independence of variables that were significantly related to aortic stiffness in elderly diabetic patients. Because age, TG, fasting glucose, BUN, creatinine, and UPCR were not normally distributed, these parameters underwent logarithmic transformations with base 10 to achieve normality. Variables that were significantly associated with cfPWV values among elderly diabetic populations were tested for independence in a linear regression analysis and then a multivariate forward stepwise regression analysis. The area under the receiver operating characteristic curve was calculated to identify the optimal adiponectin levels and significant variables associated with aortic stiffness. We used IBM SPSS Statistics Version 19.0 (SPSS Inc., Chicago, IL, USA) for data analyses. A *p*-value < 0.05 was considered statistically significant.

## 3. Results

### 3.1. The Basic Variables Compared between the Aortic Stiffness and Control Groups in Geriatric Patients with Type 2 DM

Demographic, clinical, and biochemical parameters and drugs used by the 130 elderly diabetic patients are presented in Table 1. In total, 66 (50.8%) elderly diabetic participants were included in the aortic stiffness group. Elderly diabetic participants in the aortic stiffness group had lower serum adiponectin concentrations than those in the control group (*p* < 0.001). The medications use included angiotensin-converting enzyme inhibitors (ACEi; *n* = 6; 4.6%), angiotensin receptor blockers (ARB; *n* = 59; 45.4%), β-blockers (*n* = 40; 30.8%), calcium channel blockers (CCB; *n* = 56; 43.1%), statins (*n* = 61; 46.9%), fibrate (*n* = 9; 6.9%), metformin (*n* = 56; 43.1%), sulfonylureas (*n* = 73; 56.2%), dipeptidyl peptidase 4 inhibitors (DDP-4 inhibitor; *n* = 66; 50.8%), peroxisome proliferator-activated receptor gamma agonists (PPAR-γ agonist; *n* = 7; 5.4%), and insulin (*n* = 32; 24.6%). No statistically significant differences were observed in sex, hypertension, or the use of ACEi, ARB, β-blockers, CCB, statins, fibrate, metformin, sulfonylureas, DDP-4 inhibitors, PPAR-γ agonists, or insulin between the two groups.

### 3.2. Serum Adiponectin Level Is Independent Risk Factor for Aortic Stiffness in Geriatric Patients with Type 2 DM

The unadjusted and multivariate logistic regression analyses of the parameters significantly associated with aortic stiffness, including age, waist circumference, SBP, DBP, TG, creatinine, eGFR, UPCR, and adiponectin levels are presented in Table 2. The unadjusted serum adiponectin concentrations with aortic stiffness revealed that for every 1 μg/mL that adiponectin increased, the risk of aortic stiffness in elderly diabetic participants decreased by 5.5% (odds ratio (OR): 0.945, 95% confidence interval (CI): 0.914–0.977, *p* = 0.001). Multivariate logistic regression analysis adjusted for sex and BMI showed a 5.6% decrease in the risk of aortic stiffness (OR: 0.944, 95% CI: 0.912–0.976, *p* = 0.001) for every 1 μg/mL adiponectin elevation (Model 1). After multivariate logistic regression analysis with Model 1 adjusted for fasting glucose, glycated hemoglobin, total cholesterol, and LDL-C, every 1 μg/mL adiponectin elevation showed a 5.7% drop in the risk of aortic stiffness (OR: 0.943, 95% CI: 0.911–0.976, *p* = 0.001) in Model 2. Multivariate logistic regression analysis using Model 2 adjusted for all drugs used in this study also represented a decreased risk of aortic stiffness of 7.0% (OR: 0.930, 95% CI: 0.893–0.968, *p* < 0.001) for every 1 μg/mL adiponectin elevation (Model 3). Multivariate logistic regression analysis using Model 3 adjusted for age, waist circumference, SBP, DBP, TG, creatinine, eGFR, UPCR, and adiponectin in this study also represented a decreased risk of aortic stiffness of 6.1% (OR: 0.939, 95% CI: 0.898–0.981, *p* = 0.005) for every 1 μg/mL adiponectin elevation (Model 4). Each of these analyses acknowledged that the serum adiponectin level had an inverse association with aortic stiffness in elderly diabetic participants. By using the ROC curve, the best cut-off serum value of adiponectin that could be used to predict aortic stiffness in elderly diabetic participants was 23.7 µg/mL, with the area under the receiver operating characteristic curve of 0.662 (95% CI 0.574–0.743, *p* = 0.0006), sensitivity of 48.5%, and specificity of 82.8%. The diagnostic performance of adiponectin, waist circumference, SBP, DBP, TG, and eGFR as shown in Figure 1 and Table 3.

### 3.3. Correlation between cfPWV and Clinical Variables in Geriatric Patients with Type 2 DM

Analysis of cfPWV values with variables in elderly diabetic participants via univariate and multivariate linear analyses is shown in Table 4. Waist circumference (*r* = 0.173, *p* = 0.049), SBP (*r* = 0.286, *p* = 0.001), DBP (*r* = 0.236, *p* = 0.007), logarithmically transformed TG (log-TG, *r* = 0.269, *p* = 0.002), log-creatinine (*r* = 0.207, *p* = 0.018), and log-UPCR (*r* = 0.174, *p* = 0.048) were positively correlated with cfPWV values, while eGFR (*r* = −0.215, *p* = 0.014), and adiponectin (*r* = −0.327, *p* < 0.001) had negative correlations. Multivariate forward stepwise linear regression analysis of the variables significantly associated with cfPWV values revealed that SBP (β = 0.231, adjusted R^2^ change = 0.056, *p* = 0.005), log-TG (β = 0.166, adjusted R^2^ change = 0.019, *p* = 0.048), and adiponectin (β = −0.256, adjusted R^2^ change = 0.100, *p* = 0.003) were independent parameters of cfPWV values in elderly diabetic participants.

## 4. Discussion

Our cross-sectional study showed that serum adiponectin concentration was inversely associated with cfPWV values in elderly diabetic participants, while SBP and log-TG were positively correlated. Hypoadiponectinemia was associated with aortic stiffness in elderly diabetic participants after adjusting for covariates.

Arterial stiffness, a degenerative process predominantly affecting the extracellular matrix of elastic arteries under the influence of aging and other risks, is a marker of CVD events independent of the traditional atherosclerosis risk factors [18]. Noninvasive cfPWV is currently used as the gold standard for determining aortic stiffness in clinical practice [19]. Previous studies found that age, BP, obesity, dyslipidemia, renal function, and the use of anti-hypertensive medications had a significant association with aortic stiffness [20,21,22]. Of these risk factors, aortic stiffness was principally determined by age and BP, which may account for up to 70% of its variance [20,23,24]. In our aged population, aortic stiffness was aggravated by functional and structural vasculature changes characterized by impaired endothelial function, reduced vascular distensibility, and increased vascular wall thickness. In addition, aging-related vascular changes are precipitated by arterial hypertension [25].

A previous study revealed that increased aortic stiffness was a major predictor of CVD occurrence and mortality in patients with type 2 DM [26]. DM patients have elevated aortic stiffness compared with subjects without DM, and higher aortic stiffness is correlated with the development of macro- and microvascular complications in this population [27]. Hyperlipidemia is an established marker of endothelial dysfunction and CVD risk in patients with DM [28], and a prospective, community-based study revealed that hypertriglyceridemia had a significant positive association with elevated cfPWV, similar to the findings in our study [29]. Our previous study also demonstrated that age and SBP were positively associated with cfPWV in middle-aged type 2 DM populations, while circulating adiponectin was negatively correlated [11]. However, in elderly diabetic persons, the present study demonstrated that age was not significant, and only SBP, log-triglyceride and hypoadiponectinemia were still associated with cfPWV values in elderly diabetic participants. This indicates that serum adiponectin is a good marker for this population beyond traditional risk factors.

Adiponectin, which binds to distinct G protein-coupled receptors AdipoR1 and AdipoR2, acts as an insulin sensitizer and displays antidiabetic effects [30]. Hypoadiponectinemia has been noted in coronary atherosclerosis and acute coronary syndrome populations and has been confirmed by angiography [31]. In type 2 DM patients with obesity and documented coronary atherosclerosis, circulating adiponectin levels are even lower [32]. In cases of inflammatory and oxidative vascular injury, decreased serum adiponectin levels lead to vascular function impairment and enhance cfPWV elevation [33]. Previous studies also support that hypoadiponectinemia is correlated with arterial stiffness in hypertensive populations, and a 4.6-year follow-up longitudinal study including 240 patients revealed that baseline adiponectin was an independent predictor of aortic stiffness progression [34,35,36,37]. A very current study also revealed that hypoadiponectinemia is associated with aortic stiffness among diabetic patients with stage 3–5 chronic kidney disease [38]. The mechanism of serum adiponectin that results in central arterial stiffness may be attributable to several cardiovascular and metabolic factors. Adiponectin can enhance the anti-inflammatory response, stimulate fatty acid oxidation, and improve the vasodilatory effect of endothelial nitric oxide synthase (eNOS) via activation of the intracellular protein adaptor protein phosphotyrosine interacting with pleckstrin homology domain and leucine zipper 1 (APPL1) [39]. The abovementioned modulating vascular remodeling properties can prevent the progression of arterial stiffness. In addition, adiponectin has anti-inflammatory effects that inhibit the activation of nuclear factor-kappa B in the vascular endothelium [40]. By activating eNOS and suppressing inducible NOS activity in the vasculature to limit hyperlipidemic vessel injury, adiponectin also improves endothelial function via its anti-atherosclerotic mechanism to alleviate aortic stiffness [41]. In this study, our data revealed a decreased risk of aortic stiffness of 6.0% for every 1 ng/mL adiponectin increase. After adjusting for various confounding factors in the multivariable forward stepwise linear regression analysis, circulating adiponectin level was negatively correlated with cfPWV in elder type 2 DM subjects, suggesting that a lower circulating adiponectin level was a predictor of arterial stiffness in these populations, independent of well-known CVD risk factors.

There are some limitations to our research. Firstly, this was a cross-sectional study with a limited sample size, and all patients were recruited from the same hospital; therefore, our cohort may not truly represent the wider population, and further longitudinal research is necessary before causality can be established. Secondly, there are multiple isoforms of serum adiponectin, and high-molecular-weight adiponectin has the most powerful biologic effects [7]. Although we measured total serum adiponectin levels, a previous report stated that there was a strong correlation between serum total adiponectin and high-molecular-weight adiponectin in developing CVD [42]. Thirdly, lifestyles known to aggravate aortic stiffness development—including smoking, unhealthy diet, lack of physical activity, and alcohol consumption—were not assessed and may limit the study’s predictive power [34,43,44]. Fourthly, evidence on the ethnic disparity in aortic stiffness has accumulated. A study in Singapore had revealed differences in aortic stiffness via cfPWV among the multi-ethnic type 2 DM Asian population [45]. However, there was no definite cfPWV value for defining aortic stiffness in the Asian population. Additionally, although cfPWV is age-dependent, there is no age-specific cfPWV available in Taiwanese. Therefore, we used the definition of aortic stiffness as the cfPWV values >10 m/s regardless of age in our study according to the European guidelines as another Asian population study used before [46]. Future studies may provide further information if aortic stiffness of the Asian population is available. Lastly, several medications, such as anti-hypertensives, statins, and oral anti-diabetics, acting on the dynamic component of aortic stiffness or the structural component in arterial wall remodeling, have influenced aortic stiffness [47]; however, our study demonstrated that ACEi, ARB, β-blockers, CCB, statin, fibrates, and anti-diabetic medications had no impact on aortic stiffness in elder diabetic patients, likely due to the high frequency of comorbidities in the studied cohort. Further research is needed to elucidate the impact of the abovementioned medications on aortic stiffness and long-term follow-up in this population.

## 5. Conclusions

Hypoadiponectinemia is positively associated with cfPWV values and is recognized as a consistent and significant parameter associated with aortic stiffness in elderly diabetic persons.

## Figures and Tables

**Figure 1 ijerph-19-03062-f001:**
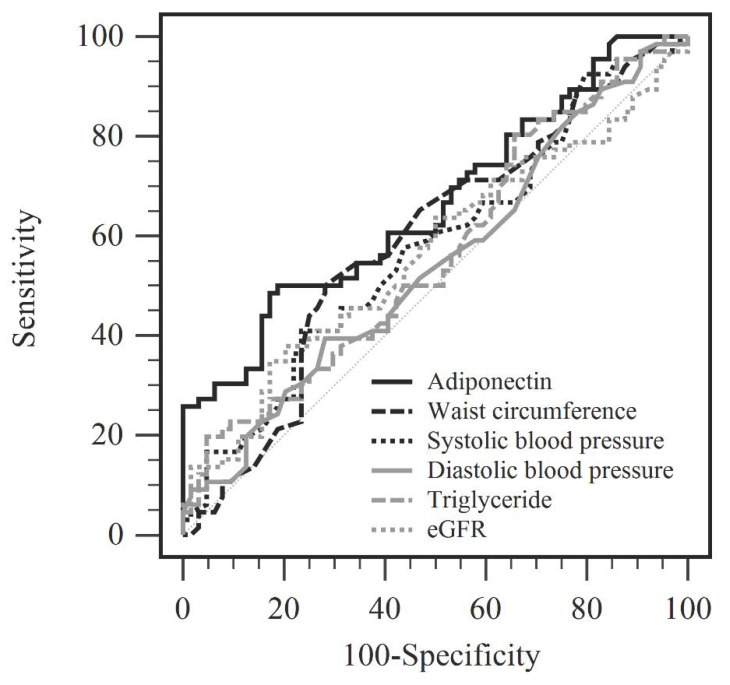
Receiver operating characteristic curve of adiponectin, waist circumference, systolic blood pressure, diastolic blood pressure, triglyceride, and eGFR on the prediction of aortic stiffness in geriatric diabetic patients.

**Table 1 ijerph-19-03062-t001:** Comparisons of clinical characteristics between aortic stiffness and control group in geriatric diabetic patients.

Variables	All Participants (*n* = 130)	Control Group (*n* = 64)	Aortic Stiffness Group (*n* = 66)	*p*-Value
Age (years)	72.00 (68.00–78.00)	70.50 (67.00–76.75)	73.50 (68.00–78.25)	0.106
Height (cm)	159.40 ± 8.94	159.26 ± 8.49	159.55 ± 9.43	0.855
Body weight (kg)	68.66 ± 13.10	68.23 ± 13.32	69.07 ± 12.97	0.713
Body mass index (kg/m^2^)	26.87 ± 3.72	26.77 ± 4.04	26.97 ± 3.41	0.757
Waist circumference (cm)	91.49 ± 9.99	90.16 ± 10.60	92.77 ± 9.26	0.137
cfPWV (m/s)	10.95 ± 3.61	8.30 ± 1.30	13.51 ± 3.26	<0.001 *
SBP (mmHg)	147.50 ± 19.34	144.52 ± 17.91	150.39 ± 20.34	0.083
DBP (mmHg)	80.78 ± 10.69	79.73 ± 10.14	81.79 ± 11.18	0.275
Total cholesterol (mg/dL)	153.14 ± 29.43	154.02 ± 29.90	152.29 ± 29.18	0.739
Triglyceride (mg/dL)	118.00 (91.00–158.25)	118.00 (81.75–151.50)	118.50 (95.00–164.50)	0.215
LDL-C (mg/dL)	88.32 ± 25.45	87.50 ± 25.80	89.12 ± 25.28	0.718
Fasting glucose (mg/dL)	131.00 (113.75–167.75)	131.00 (111.00–170.75)	130.50 (115.25–161.50)	0.909
HbA1c (%)	7.05 (6.50–8.53)	7.00 (6.43–8.55)	7.10 (6.60–8.53)	0.137
Blood urea nitrogen (mg/dL)	20.00 (16.00–30.00)	19.00 (16.00–27.00)	20.50 (17.00–31.00)	0.305
Creatinine (mg/dL)	1.15 (0.80–1.70)	1.10 (0.80–1.58)	1.30 (0.88–1.80)	0.146
eGFR (mL/min)	61.70 ± 28.81	64.50 ± 26.34	58.99 ± 30.98	0.277
UPCR (mg/g)	93.66 (0.00–309.49)	82.51 (0.00–192.88)	97.24 (0.00–396.38)	0.277
Adiponectin (μg/mL)	31.82 ± 13.79	36.11 ± 15.41	27.67 ± 10.57	<0.001 *
Female, *n* (%)	57 (43.8)	28 (43.8)	29 (43.9)	0.983
Hypertension, *n* (%)	93 (71.5)	46 (71.9)	47 (71.2)	0.933
ACE inhibitor use, *n* (%)	6 (4.6)	3 (4.7)	3 (4.5)	0.969
ARB use, *n* (%)	59 (45.4)	28 (43.8)	31 (47.0)	0.712
β-blocker use, *n* (%)	40 (30.8)	16 (25.0)	24 (36.4)	0.160
CCB use, *n* (%)	56 (43.1)	26 (40.6)	30 (45.5)	0.578
Statin use, *n* (%)	61 (46.9)	31 (48.4)	30 (45.5)	0.733
Fibrate use, *n* (%)	9 (6.9)	5 (7.8)	4 (6.1)	0.694
Metformin use, *n* (%)	56 (43.1)	27 (42.2)	29 (43.9)	0.840
Sulfonylurea use, *n* (%)	73 (56.2)	39 (60.9)	34 (51.5)	0.279
DDP-4 inhibitor use, *n* (%)	66 (50.8)	36 (56.3)	30 (45.5)	0.218
PPAR-γ agonist use, *n* (%)	7 (5.4)	5 (7.8)	2 (3.0)	0.227
Insulin use, *n* (%)	32 (24.6)	19 (29.7)	13 (19.7)	0.186

Values for continuous variables are given as the mean ± standard deviation and tested by Student’s *t*-test; variables not normally distributed are given as the median and interquartile range and tested by Mann–Whitney U test; values are presented as number (%) and analysis was performed using the chi-square test. Abbreviations: cfPWV, carotid–femoral pulse wave velocity; SBP, systolic blood pressure; DBP, diastolic blood pressure; LDL-C, low-density lipoprotein cholesterol; HbA1c, glycated hemoglobin; eGFR, estimated glomerular filtration rate; UPCR, urine protein-to-creatinine ratio; ACE, angiotensin-converting enzyme; ARB, angiotensin receptor blocker; CCB, calcium channel blocker; DDP-4, dipeptidyl peptidase 4; PPAR-γ, peroxisome proliferator-activated receptor gamma. * *p* < 0.05 was considered statistically significant.

**Table 2 ijerph-19-03062-t002:** Variables associated with aortic stiffness by multivariable logistic regression analysis among the 130 geriatric diabetic patients.

Variables	Unadjusted		Model 1		Model 2		Model 3		Model 4	
	OR (95% CI)	*p* Value	OR (95% CI)	*p* Value	OR (95% CI)	*p* Value	OR (95% CI)	*p* Value	OR (95% CI)	*p* Value
Adiponectin, 1 μg/mL	0.945 (0.914–0.977)	0.001 *	0.944 (0.912–0.976)	0.001 *	0.943 (0.911–0.976)	0.001 *	0.930 (0.893–0.968)	<0.001 *	0.939 (0.898–0.981)	0.005 *
Age, 1 year	1.041 (0.987–1.098)	0.144	1.044 (0.989–1.102)	0.121	1.053 (0.995–1.114)	0.074	1.040 (0.977–1.106)	0.218	1.604 (0.979–1.157)	0.146
Waist circumference, 1 cm	1.027 (0.991–1.065)	0.140	1.068 (1.004–1.136)	0.037 *	1.069 (1.005–1.138)	0.033 *	1.090 (1.016–1.169)	0.016 *	1.064 (0.981–1.155)	0.133
Systolic blood pressure, 1 mmHg	1.016 (0.998–1.035)	0.086	1.017 (0.998–1.036)	0.089	1.018 (0.999–1.038)	0.069	1.015 (0.994–1.037)	0.168	1.022 (0.985–1.061)	0.239
Diastolic blood pressure, 1 mmHg	1.018 (0.986–1.052)	0.274	1.018 (0.985–1.053)	0.287	1.020 (0.986–1.054)	0.258	1.020 (0.983–1.058)	0.299	0.977 (0.918–1.040)	0.466
Triglyceride, 1 mg/dL	1.005 (0.999–1.010)	0.087	1.005 (0.999–1.010)	0.087	1.009 (1.002–1.016)	0.009 *	1.012 (1.004–1.020)	0.004 *	1.008 (0.999–1.017)	0.071
Creatinine, 1 mg/dL	1.511 (0.941–2.428)	0.088	1.528 (0.947–2.464)	0.082	1.607 (0.976–2.647)	0.062	1.655 (0.894–3.062)	0.109	1.441 (0.419–4.960)	0.562
eGFR, 1 mL/min	0.993 (0.981–1.005)	0.275	0.993 (0.981–1.005)	0.277	0.992 (0.979–1.004)	0.195	0.990 (0.976–1.005)	0.211	1.012 (0.979–1.046)	0.487
UPCR, 1 mg/g	1.000 (1.000–1.001)	0.400	1.000 (1.000–1.001)	0.412	1.000 (1.000–1.001)	0.340	1.000 (1.000–1.001)	0.368	1.000 (0.999–1.001)	0.811

Multivariate logistic regression analysis was applied to process the data. Model 1 is adjusted for gender and BMI. Model 2 is adjusted for the Model 1 variables and for fasting glucose, glycated hemoglobin, total cholesterol, and LDL-C. Model 3 is adjusted for the Model 2 variables and all drugs, such as ACE inhibitor, ARB, β-blocker, CCB, statin, fibrate, metformin, sulfonylureas, DDP-4 inhibitor, PPAR-γ agonist, and insulin. Model 4 is adjusted for the Model 3 variables and age, waist circumference, systolic blood pressure, diastolic blood pressure, triglyceride, creatinine, eGFR, UPCR, and adiponectin. OR, odds ratio; CI, confidence interval; BMI, body mass index; SBP, systolic blood pressure; DBP, diastolic blood pressure; LDL-C, low-density lipoprotein cholesterol; eGFR, estimated glomerular filtration rate; UPCR, urine protein-to-creatinine ratio; ACE, angiotensin-converting enzyme; ARB, angiotensin receptor blocker; CCB, calcium channel blocker; DDP-4, dipeptidyl peptidase 4; PPAR-γ, peroxisome proliferator-activated receptor gamma. * *p* < 0.05 was considered statistically significant.

**Table 3 ijerph-19-03062-t003:** The diagnostic performance of adiponectin, waist circumference, systolic blood pressure, diastolic blood pressure, triglyceride, and eGFR on aortic stiffness in geriatric diabetic patients.

Variables	AUC (95% CI)	*p* Value
Adiponectin	0.662 (0.574–0.743)	0.0006 *
Waist circumference	0.591 (0.501–0.676)	0.0732
Systolic blood pressure	0.577 (0.487–0.663)	0.1276
Diastolic blood pressure	0.543 (0.454–0.631)	0.3924
Triglyceride	0.563 (0.473–0.650)	0.2134
eGFR	0.565 (0.476–0.652)	0.1989

AUC, area under curve; CI, confidence interval; eGFR, estimated glomerular filtration rate. * *p* < 0.05 was considered statistically significant.

**Table 4 ijerph-19-03062-t004:** Correlation between carotid–femoral pulse wave velocity levels and clinical variables in geriatric diabetic patients.

Variables	Carotid–Femoral Pulse Wave Velocity (m/s)				
	Simple Regression		Multivariate Regression		
	*r*	*p* Value	Beta	Adjusted R^2^ Change	*p* Value
Female	0.070	0.432	–	–	–
Hypertension	0.049	0.578	–	–	–
Log-age (years)	0.092	0.298	–	–	–
Height (cm)	0.020	0.824	–	–	–
Body weight (kg)	0.032	0.720	–	–	–
Body mass index (kg/m^2^)	0.028	0.755	–	–	–
Waist circumference (cm)	0.173	0.049 *	–	–	–
SBP (mmHg)	0.286	0.001 *	0.231	0.056	0.005 *
DBP (mmHg)	0.236	0.007 *	–	–	–
Total cholesterol (mg/dL)	0.035	0.690	–	–	–
Log-triglyceride (mg/dL)	0.269	0.002 *	0.166	0.019	0.048 *
LDL-C (mg/dL)	−0.005	0.955	–	–	–
Log-glucose (mg/dL)	0.027	0.757	–	–	–
Log-HbA1c (%)	0.099	0.263	–	–	–
Log-BUN (mg/dL)	0.164	0.062	–	–	–
Log-creatinine (mg/dL)	0.207	0.018 *	–	–	–
eGFR (mL/min)	−0.215	0.014 *	–	–	–
Log-UPCR (mg/g)	0.174	0.048 *	–	–	–
Adiponectin (μg/mL)	−0.327	<0.001 *	−0.256	0.100	0.003 *

Data on age and triglyceride, glucose, HbA1c, BUN, creatinine, and UPCR levels showed skewed distribution and therefore were log-transformed before analysis. Analysis of data was performed using univariate linear regression analysis or multivariate stepwise linear regression analysis (adapted factors were waist circumference, SBP, DBP, log-triglyceride, log-creatinine, eGFR, log-UPCR, and adiponectin). Abbreviations: SBP, systolic blood pressure; DBP, diastolic blood pressure; LDL-C, low-density lipoprotein cholesterol; HbA1c, glycated hemoglobin; BUN, blood urea nitrogen; eGFR, estimated glomerular filtration rate; UPCR, urine protein-to-creatinine ratio. * *p* < 0.05 was considered statistically significant.

## Data Availability

The data presented in this study are available on request from the corresponding author.

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
