# Peer review of "Negative Correlation of Serum Adiponectin Level with Aortic Stiffness in Elderly Diabetic Persons"

_ijerph, 2022, doi:10.3390/ijerph19053062_

Round 1

Reviewer 1 Report

he manuscript “Negative Correlation of Serum Adiponectin Level with Aortic Stiffness in Elderly Diabetic Persons” The topic addressed here is relevant for general and specialist medical science and valuable in clinical and in health political strategies areas. The manuscript is well written; however, some methodological clarifications are needed it.

Methods

Explain how medications can affect the aortic stiffness

The control and study groups are not clearly explained in Methods.

Line 140. The definition of hypertension required a reference

Results

Line  126. “The baseline variables compared between the aortic stiffness and control groups in geriatric patients with type 2 DM” the term baseline is confusing here because it is generalized used to refer to the value before any treatment for longitudinal studies. Please consider changing it.

In section 3.2, “Serum adiponectin level is an independent risk factor for aortic stiffness in geriatric patients with type 2 DM,” the authors refer to Model 1, Model 2, and Model 3. These models appear to be differentiated by the variable’s adjustments. Please clarify the used models in Methods-Data analysis.

Line 180 “Subgroup analysis of cfPWV….” It is not clear why the authors used a subgroup for correlations instead of the whole sample

Discussion

The interpretation of the result from ROC curves is missing.

References

Up-to-date examples of similar studies should be included. i.e.

Angiology 2020 https://doi.org/10.1177/0003319720927203

Vascular 2021 https://doi.org/10.1177/17085381211007602

Author Response

Response to Reviewer 1: 

We thank the reviewers for their helpful comments. We have revised the manuscript according to the reviewers' comments/suggestions, and have provided an itemized Response in the following sections.

The manuscript "Negative Correlation of Serum Adiponectin Level with Aortic Stiffness in Elderly Diabetic Persons" The topic addressed here is relevant for general and specialist medical science and valuable in clinical and in health political strategies areas. The manuscript is well written; however, some methodological clarifications are needed it.

Methods

Explain how medications can affect the aortic stiffness

Response: Thanks for pointing this out. According to the literature, several drugs have been shown to influence aortic stiffness, such as anti-hypertensives, statins, or oral anti-diabetics. Therefore, we analyzed these medications of our study subjects as the clinical characteristics between aortic stiffness and control group in geriatric diabetic patients. We had mentioned and added it in the Limitation part and cited reference 47. We had added this information in the Limitation part of the manuscript as " Several medications, such as anti-hypertensives, statins, and oral anti-diabetics, acting on the dynamic component of aortic stiffness or the structural component in arterial wall remodeling, have influenced aortic stiffness." Further researches are needed to elucidate the impact of the abovementioned medications on aortic stiffness in this population.

The control and study groups are not clearly explained in Methods.

Response: Thanks for pointing this out. We apologize that the definition of our study's control group is unclear. We added the following information in the Materials and Methods parts 2.4 Aortic stiffness by cfPWV measurements of the manuscript to define the control group in geriatric diabetic patients without aortic stiffness. "We defined aortic stiffness as the cfPWV values > 10 m/s, , and those whose values were ≤ 10 m/s were included in the control group in our study according to the guidelines published by the European Society of Cardiology and the European Society of Hypertension."

Line 140. The definition of hypertension required a reference

Response: Thanks for pointing this out. We had added the reference 13 of “according to 2018 European Society of Cardiology and the European Society of Hypertension guidelines for the management of arterial hypertension." to the Method part of the manuscript. Thanks for your comments.

Results

Line 126. "The baseline variables compared between the aortic stiffness and control groups in geriatric patients with type 2 DM" the term baseline is confusing here because it is generalized used to refer to the value before any treatment for longitudinal studies. Please consider changing it.

Response: Thanks for pointing this out. We agree with the reviewer's comment and have changed the word "baseline" to "basic" to avoid confusion.

In section 3.2, "Serum adiponectin level is an independent risk factor for aortic stiffness in geriatric patients with type 2 DM," the authors refer to Model 1, Model 2, and Model 3. These models appear to be differentiated by the variable's adjustments. Please clarify the used models in Methods-Data analysis.

Response: We describes more detail in Statistical analysis and in Table 2. Multivariate logistic regression analysis was performed to test the independence of variables that were significantly related to aortic stiffness in elderly diabetic patients by adjusted for age, gender, waist circumference, BMI, SBP, DBP, fasting glucose, glycated hemoglobin, eGFR, UPCR, total cholesterol, triglyceride, LDL-C and drugs used in this study in different models. Thanks for your comments.

Line 180 "Subgroup analysis of cfPWV…." It is not clear why the authors used a subgroup for correlations instead of the whole sample

Response: Thanks for pointing this out. We apologize that the statement of subgroup analysis is incorrect and we use whole sample. We corrected it as “Analysis of cfPWV values with variables in elderly diabetic participants via univariate and multivariate linear analyses is shown in Table 3.” Thanks for your comments.

Discussion

The interpretation of the result from ROC curves is missing.

Response: Thanks for pointing this out. We added the data as “By using the ROC curve, the best cut-off serum value of adiponectin that can be used to predict aortic stiffness in elderly diabetic participants was 23.7 µg/mL, with the area under the receiver operating characteristic curve of 0.662 (95% CI 0.574–0.743, p = 0.0006), sensitivity of 48.5%, and specificity of 82.8% as shown in Figure 1.” Thanks for your comments.

References

Up-to-date examples of similar studies should be included. i.e.

Angiology 2020 https://doi.org/10.1177/0003319720927203 and Vascular 2021 https://doi.org/10.1177/17085381211007602

Response: Thanks for the suggestion. We had added this information to the fourth paragraph of our Discussion part and cited these two excellent studies to our references (references 37 and 38) as your suggestion.

Reviewer 2 Report

The Authors present an interesting study about the effect of low serum adiponectin levels on aortic stiffness in geriatric diabetic patients. The results suggest that serum adiponectin is negatively correlated with carotid-femoral pulse wave velocity and is an independent predictor of aortic stiffness in elderly diabetic persons.
The analysis is generally well-conceived and executed, and the Manuscript is well written and easy to follow. However, the sample size for the association analyses is very limited (130 participants) and a bigger samples size would be required to validate the results.
How can the Authors prove that age is not significant on aortic stiffness? The age window of the participants is very limited to state this kind of statement.

Author Response

We thank the reviewers for their helpful comments. We have revised the manuscript according to the reviewers' comments/suggestions and provided an itemized Response in the following sections.

The Authors present an interesting study about the effect of low serum adiponectin levels on aortic stiffness in geriatric diabetic patients. The results suggest that serum adiponectin is negatively correlated with carotid-femoral pulse wave velocity and is an independent predictor of aortic stiffness in elderly diabetic persons.
The analysis is generally well-conceived and executed, and the manuscript is well written and easy to follow. However, the sample size for the association analyses is very limited (130 participants) and a bigger samples size would be required to validate the results.
Response: Thanks for the reminder. We had stated this in the 5th paragraph of our Discussion part as our study limitation as "Firstly, this was a cross-sectional study with a limited sample size and all patients were recruited from the same hospital. Therefore, our cohort may not truly represent the wider population; further longitudinal researches are necessary before causality can be established."

How can the Authors prove that age is not significant on aortic stiffness? The age window of the participants is very limited to state this kind of statement.

Response: Thanks for pointing this out. Aortic stiffness measured by cfPWV is age-dependent, and accumulating studies have reported the impact of age on arterial stiffness. Our previous study had demonstrated that age and SBP were positively associated with cfPWV in middle-aged type 2 DM populations while circulating adiponectin was negatively correlated [Shih CH, Hsu BG, Hou JS, Wu DA, Subeq YM. Association of Low Serum Adiponectin Levels with Aortic Arterial Stiffness in Patients with Type 2 Diabetes. J Clin Med. 2019 Jun 21;8(6):887.]. However, our present study focused on elderly diabetic persons; the result demonstrated that age was not statistically significant between aortic stiffness and the control group. After multivariate forward stepwise linear regression analysis of the variables significantly associated with cfPWV values, only systolic blood pressure, log-triglyceride and adiponectin were still associated with cfPWV values in elderly diabetic persons. So, we concluded that age is not significant on aortic stiffness in elderly diabetic patients. This information had written in the third paragraph of the Discussion part of the manuscript.

Reviewer 3 Report

In this study, the authors reported a negative correlation between serum adiponectin and carotid-femoral pulse wave velocity-defined aortic stiffness. Moreover, they showed that adiponectin level was independent predicting aortic stiffness in results of multivariate forward stepwise linear regression analysis. The finding seems interesting however there are some concerns.

  1. The authors used European guideline to define the presence of aortic stiffness. However, the study population was Asian. Besides, carotid-femoral pulse wave velocity is age-dependent but the age factor was not taken in their definition of aortic stiffness.
  2. The adiponectin levels are in different range from other publications. For example,

Yeli Wang et al. reports “Plasma adiponectin levels and type 2 diabetes risk” in a Singapore Chinese population. Their results showed the median (interquartile) concentration of adiponectin was 6.7 (5.2–8.3) µg/mL in men and 8.1 (6.4–10.5) µg/mL in women.

  1. The authors used enzyme assay to measure adiponectin levels. What are the within-assay and between-assay coefficients of variation?

Author Response

We thank the reviewers for their helpful comments. We have revised the manuscript according to the reviewers' comments/suggestions and provided an itemized Response in the following sections.

In this study, the authors reported a negative correlation between serum adiponectin and carotid-femoral pulse wave velocity-defined aortic stiffness. Moreover, they showed that adiponectin level was independent predicting aortic stiffness in results of multivariate forward stepwise linear regression analysis. The finding seems interesting however there are some concerns.

1. The authors used European guideline to define the presence of aortic stiffness. However, the study population was Asian. Besides, carotid-femoral pulse wave velocity is age-dependent but the age factor was not taken in their definition of aortic stiffness.

Response: Thanks for pointing this out. Evidence on the ethnic disparity in arterial stiffness has begun to accumulate. A study in Singapore had revealed the different central arterial stiffness via carotid-femoral pulse wave velocity (cfPWV) among the multi-ethnic T2DM Asian population [Zhang X, Liu JJ, Sum CF, Ying YL, Tavintharan S, Ng XW, Low S, Lee SB, Tang WE, Lim SC. Ethnic disparity in central arterial stiffness and its determinants among Asians with type 2 diabetes. Atherosclerosis. 2015 Sep;242(1):22-28]. However, there was no definite cfPWV value to define the aortic stiffness in the Asain population. Besides, although cfPWV is age-dependent but there is no age-specific cfPWV available in Taiwanese. Therefore, we used the definition of aortic stiffness as the cfPWV values > 10 m/s regardless of age in our study according to the guidelines published by the European Society of Cardiology and the European Society of Hypertension as another Asain population study [Moh MC, Sum CF, Tavintharan S, Ang K, Lee SBM, Tang WE, Lim SC. Baseline predictors of aortic stiffness progression among multi-ethnic Asians with type 2 diabetes. Atherosclerosis. 2017 May;260:102-109.]. We had added this information to our limitation part of the Discussion of the manuscript as "Fourthly, evidence on the ethnic disparity in aortic stiffness has accumulated. A study in Singapore had revealed the different aortic stiffness via carotid-femoral pulse wave velocity (cfPWV) among the multi-ethnic T2DM Asian population. However, there was no definite cfPWV value to define the aortic stiffness in the Asain population. Besides, although cfPWV is age-dependent, there is no age-specific cfPWV available in Taiwanese. Therefore, we used the definition of aortic stiffness as the cfPWV values > 10 m/s regardless of age in our study according to the European guidelines as another Asain population study before. Future studies may provide further information if aortic stiffness of the Asian population is available."

2. The adiponectin levels are in different range from other publications. For example,

Yeli Wang et al. reports "Plasma adiponectin levels and type 2 diabetes risk" in a Singapore Chinese population. Their results showed the median (interquartile) concentration of adiponectin was 6.7 (5.2–8.3) µg/mL in men and 8.1 (6.4–10.5) µg/mL in women.
Response: Thanks for pointing this out. We apologize that we typing error of the unit of adiponectin and the unit is µg/mL. Many studies noted the medium adiponectin level in DM subjects are higher than the study of Wang et al. such as Nielsen et al. (median plasma adiponectin of 28.9 μg/mL (4th quartile), 19.2 μg/mL (3rd quartile), 13.9 μg/mL (2nd quartile), and 9.2 μg/mL (1st quartile)) or Liao et al. (22.13 µg/mL) and noted in the blow references. Thanks for your comments.

Nielsen MB, Çolak Y, Benn M, Nordestgaard BG. Low Plasma Adiponectin in Risk of Type 2 Diabetes: Observational Analysis and One- and Two-Sample Mendelian Randomization Analyses in 756,219 Individuals. Diabetes. 2021 Nov;70(11):2694-2705.

Liao PJ, Ting MK, Wu IW, Chen SW, Yang NI, Hsu KH. Higher Leptin-to-Adiponectin Ratio Strengthens the Association Between Body Measurements and Occurrence of Type 2 Diabetes Mellitus. Front Public Health. 2021 Jul 23;9:678681.

3. The authors used enzyme assay to measure adiponectin levels. What are the within-assay and between-assay coefficients of variation?

Response: The intra-assay and inter-assay coefficients of variation in the measurement for adiponectin was 6.4% and 7.3%, respectively. We added it in the Biochemical investigations. Thanks for your comments.

Round 2

Reviewer 1 Report

The authors answered all my doubt properly and attended to all my suggestions.

Author Response

Thanks for your comments.